# Study of Mechanical Properties of an Eco-Friendly Concrete Containing Recycled Carbon Fiber Reinforced Polymer and Recycled Aggregate

**DOI:** 10.3390/ma13204592

**Published:** 2020-10-15

**Authors:** Chen Xiong, Tianhao Lan, Qiangsheng Li, Haodao Li, Wujian Long

**Affiliations:** Guangdong Provincial Key Laboratory of Durability for Marine Civil Engineering, Shenzhen University, Shenzhen 518060, China; xiongchen@szu.edu.cn (C.X.); lantianhao2018@email.szu.edu.cn (T.L.); liqiangsheng@email.szu.edu.cn (Q.L.); lihaodao2017@email.szu.edu.cn (H.L.)

**Keywords:** recycled carbon fiber reinforced polymer, recycled aggregate, mechanical properties, flexural performance, eco-friendly concrete

## Abstract

This study investigates the feasibility of collaborative use of recycled carbon fiber reinforced polymer (RCFRP) fibers and recycled aggregate (RA) in concrete, which is called RCFRP fiber reinforced RA concrete (RFRAC). The mechanical properties of the composite were studied through experimental investigation, considering different RCFRP fiber contents (0%, 0.5%, 1.0%, and 1.5% by volume) and different RA replacement rates (0%, 10%, 20%, and 30% by volume). Specifically, ten different mixes were designed to explore the flowability and compressive and flexural strengths of the proposed composite. Experimental results indicated that the addition of RCFRP fibers and RA had a relatively small influence on the compressive strength of concrete (less than 5%). Moreover, the addition of RA slightly decreased the flexural strength of concrete, while the addition of RCFRP fibers could significantly improve the flexural performance. For example, the flexural strength of RA concrete with 1.5% RCFRP fiber addition increased by 32.7%. Considering the good flexural properties of the composite and its potential in reducing waste CFRP and construction solid waste, the proposed RFRAC is promising for use in civil concrete structures with high flexural performance requirements.

## 1. Introduction

With the growing process of urbanization, a huge amount of construction waste is discarded annually, which poses a great threat to the ecological environment. Research of Huang et al. [1] shows that approximately 30% of the world’s municipal solid waste is produced in China, 40% of which is construction and demolition waste. Recycled aggregate (RA) concrete is an environmentally-friendly construction material and can be used in various civil structures [2,3,4,5,6,7,8,9,10]. Therefore, extensive use of RA concrete materials contributes to economic and environmental sustainability in the long run [2].

Many researchers have carried out various attempts to extend the application of RA in civil infrastructures, such as sidewalks, highways, tunnels, and building foundations [3,4,5,6,7,8]. RA concrete generally exhibits relatively poor mechanical and durability properties [11]. The work of Evangelista et al. [12] indicated that with the increases of RA replacement, the tensile splitting strength and modulus of elasticity decreased. The study of Kou et al. [13] showed that the drying shrinkage and creep of concrete increased with the increase of RA content, while the compressive strength and durability decreased with the increase of RA content. The addition of fibers is an effective way to improve the mechanical properties of RA concrete. The research of Carneiro et al. showed that steel fibers can improve the strength and fracture performance of RA concrete [14]. The research of Akça et al. indicated that polypropylene fibers can improve the flexural and splitting strengths of RA concrete [15]. Prasad et al. found that the addition of glass fiber can improve the flexural/compressive strength, energy absorption capacity, and ductility of RA concrete [16]. The addition of fibers in concrete can also improve the seismic performance of concrete structures. For example, Xu et al. [17] investigated the seismic performance of concrete columns with the addition of steel fibers. The results show that the existence of steel fiber could prevent the cracked concrete from spalling and delay the bulking of longitudinal reinforcement. Huang et al. [18] studied the seismic performance of concrete columns with steel–polypropylene hybrid fibers. The columns exhibited a notable synergetic effect in terms of ductility and energy dissipation capacity, particularly for columns with a higher axial compression ratio. The improvement of seismic performance are primarily attributed to the bridging action provided by fibers [19]. It is believed that other fibers, such as carbon fibers, can also be adopted to improve the seismic performance of concrete structures.

Carbon fiber reinforced polymer (CFRP) is a widely used material in automobile, aviation, aerospace, sports equipment, and wind energy industries, owing to its high strength, relatively low density, good dimensional stability, and good corrosion resistance [20]. Since CFRP is widely used in industry and has been used for many years, a large amount of CFRP waste will be generated from CFRP products beyond their service life. According to the work of Meng [21], an estimated 62,000 tonnes of CFRP in production and end-of-life waste will be generated by 2020. Lefeuvre et al. estimated that by 2050, the aeronautics and wind power sectors worldwide will generate 527,374 and 483,000 tonnes of CFRP waste products, respectively [22,23]. Traditional disposal methods for carbon fibers include waste disposal or combustion. Nevertheless, these disposal solutions can pose a significant impact on the environment, such as land use, groundwater pollution, or harmful gas emissions [24]. Considering the environmental impact of waste CFRP and its potential economic value of recycling, to date, researchers have carried out a series of research on CFRP recycling, which are as follows. (1) Pyrolysis recovery technology: under certain temperature conditions, the matrix resin in CFRP composite is decomposed to realize the recovery of fibers. The advantage of this method is that the mechanical properties of carbon fibers are greatly retained, and chemical feedstock can be recovered from the resin, while the disadvantage is that environmentally hazardous off-gases are exhausted [25]. (2) Solvent decomposition: solvolysis consists of chemical treatment using a solvent to degrade resin to recover fiber. Fiber performance can be significant degraded via solvent decomposition, and some hazardous solvents may cause environmental pollution [26]. (3) Mechanical recycling: the waste CFRP sheets or plates can be cut into small fibers through a shredding machine. This method is relatively simple and has little impact on the environment. The length of short fibers obtained through mechanical recycling can be affected by the recycle machinery and the size of waste CFRP raw materials. The research of Gopalraj et al. [27] shows that the maximum length of mechanically recycled fibers is 100 mm, and the study of Mastali et al. [28] adopts short fibers with the length of 30 mm. In this paper, recycled short fibers with the length of 35 mm are adopted. Therefore, the length of mechanically recycled short fibers usually ranges from 0 to 100 mm.

To date, some exploratory works have been reported on the development of recycled carbon fiber reinforced concrete. Experimental results show that the addition of a small number of recycled carbon fibers can significantly improve the mechanical properties of concrete [28,29]. In the work of Ogi et al. [29], recycled CFRP (RCFRP) pieces with irregular shapes were adopted. The addition of RCFRP pieces in concrete improved its flexural strength and the fracture toughness. Nevertheless, owing to the irregular shape of the adopted RCFRP pieces, the tensile strength of RCFRP pieces could not be fully developed. Mastali et al. investigated the mechanical performance of concrete with the addition of recycled carbon fibers. The results reveal that both compressive strength and impact resistance of concrete can be significantly improved [28].

From the discussion above, it can be concluded that the mechanical properties (e.g., flexural and compressive strengths) of RA concrete are lower than those of conventional concrete, while these strengths can be increased by the addition of carbon or CFRP fibers. Therefore, an eco-friendly RCFRP fiber-enhanced RA concrete is proposed in this study, and the mechanical properties of the composite are examined. Specifically, the design details of different mixes and the adopted test methods are presented in Section 2. In Section 3, the flowability, compressive/flexural strengths, flexural toughness, and residual flexural load-bearing capacity of different mixes are discussed.

## 2. Materials and Test Methods

### 2.1. Materials

Ordinary Portland cement (OPC) 42.5R was adopted for all concrete mixes. The chemical composition of the selected cement is shown in Table 1, which meets the Chinese standard [30]. The sand density used in the experiment was 2.63 g/cm^3^. The particle-size distribution of sand is shown in Figure 1a, which meets the Chinese standard [31].

The nominal size of the coarse natural aggregate (NA) (Figure 2a) was between 2.36 and 19 mm. To make the experimental results more reliable, the NA was washed with water and dried in an oven, at a temperature of 60 °C ± 5 °C, and then cooled to room temperature before use.

The RA (Figure 2b) was used as the partial replacement of coarse NA in this study. The RA used in the experiment came from the Xili stone crushing factory in Nanshan district, Shenzhen, China. The RA was derived from waste road concrete (Figure 2c), and the compressive strength was between 20 and 30 MPa. After crushing the bulk concrete by the crushing machine (Figure 2d), the RA with a nominal diameter of 4.75–19 mm was screened for use. Subsequently, the RA was repeatedly washed to remove impurities and treated by the drying procedure to obtain the dried RA (Figure 2b). According to the Chinese standards [32,33], detailed information of NA and RA, including water absorption, apparent density, and crushing index, are presented in Table 2. It can be seen that RA had a high water absorption, which was 4.18 times that of NA; the apparent density of RA and NA were very close; the crushing index of RA was higher than that of NA, indicating that the RA was more fragile.

In this study, the RCFRP fibers were recovered from waste CFRP sheets with a thickness of 0.2 mm. The shredding tool shown in Figure 3a was selected to cut the waste CFRP sheets into short fibers (Figure 3b). This method only involved mechanical cutting and did not require removal of the resin in the CFRP sheets. The obtained RCFRP fibers had a regular shape, which could make good use of the tensile strength of RCFRP fibers. In addition, the RCFRP fibers could be easily dispersed in concrete, which posed a relatively small impact on the workability of the concrete. The density, tensile strength, elastic modulus, and carbon content of the waste CFRP sheets were 1.70 g/cm^3^, 4292 MPa, 236 GPa, and 92.5%, respectively. The waste CFRP sheets were cut into dimensions of 35 × 2 × 0.2 mm using the shredding machine, as shown in Figure 3a.

### 2.2. Mix Design

The objective of this experiment was to investigate the influence of RA and RCFRP fiber additions on the mechanical properties of concrete. This study maintained the absolute water amount constant for all mixes. Although the nominal water/cement ratios of different mixes were identical, the strong water absorption of RA could result in a lower actual water/cement ratio for mixes with high RA replacement rates. Even though the effects of water absorption of RA were not eliminated, the experiment of RA specimens with the same absolute water amount could still provide useful insights into the synergetic effect of RA replacement rate and water absorption of RA on the mechanical properties of RAC. According to the work of Poon et al. [34], if the recycled aggregate is used in the oven-dried state, water may move from the bulk cement matrix toward the recycled aggregate, which can result in a slightly higher compressive strength compared with that of the case with the recycled aggregate in the saturated surface-dried state. The RAC mixes were treated as the reference mixes to investigate the mechanical performance of mixes with/without RCFRP fibers. The results of these comparisons were conducted based on the same RA replacement rate, which was not significantly influenced by the water/cement ratio change induced by RA addition.

In this study, RAC mixes with the aggregate replacement rates of 0%, 10%, 20%, and 30% were investigated. To further study the performance of RAC with RCFRP fibers, the representative RA replacement rate of 20% was selected. This is because the 10% RA replacement rate was relatively small, and the effect of saving natural aggregate was not obvious in this replacement rate. Moreover, RA has the characteristic of low density, low wear resistance and compressive strength, high water absorption, and porosity [35]. Too much addition of RA may affect the durability, mechanical properties, and stability of RAC. According to the initial study conducted on RAC when the replacement rate was varied between 10 and 30% as shown in Figures 7a, 11a and 13a, the RAC mix with 30% RA replacement exhibited relatively large strength reduction compared with the mixes with 10% and 20% RA replacement rates. Therefore, the mix with an RA replacement rate of 20% was selected to further investigate the synergetic effect of RAC with RCFRP fibers. Specifically, mixes with 0% and 20% RA replacement rate and 0%, 0.5%, 1%, and 1.5% volume content of RCFRP fibers were designed. The details of the studied mixes are shown in Table 3. To improve the workability of concrete, a polycarboxylate-based high range water reducer (HRWR), with a density of 1.02 g/cm^2^, was adopted. The mass ratio of HRWR to cement was 0.194%. The content of each ingredient is given in Table 3 considering the volume change induced by RCFRP fiber addition. For example, the density of RA0RF10 was 2385 kg/m^3^, which was slightly smaller than that of RA0RF0 (2400.5 kg/m^3^).

### 2.3. Sample Preparation

The concrete preparation procedure, based on the Chinese standard [36], is illustrated in Figure 4. First, all raw materials were weighed according to the mixing ratio in Table 3. Second, NA, RA, and sand were added to the blender and stirred for 1 min. Third, cement was poured into the blender and mixed for 1 min. Forth, HRWR and water were mixed evenly, and half of the water–HRWR mixture was added to the blender and stirred for 2 min. Fifth, CFRP fibers were slowly and evenly added to avoid fiber concentration. Sixth, the remaining water–HRWR mixture was added and stirred for 2 min. Finally, the prepared mixture was poured into the 100 × 100 × 100 mm and 100 × 100 × 400 mm molds, and three specimens were made for each mix. After curing for 24 h, the samples were demolded and cured at room temperature of 20 ± 2 °C, with the humidity not less than 95%, for 28 days.

### 2.4. Test Methods

#### 2.4.1. Slump Test

The flowability of each mix was evaluated through the slump test, according to the Chinese standard [36]. The slump values ranged from 0 to 300 mm, representing the flowability of concrete.

#### 2.4.2. Compressive Strength Test

A compressive strength test was performed according to the Chinese standard [37]. Three cubic specimens, with dimensions of 100 × 100 × 100 mm, were made for each mixture. In this study, a testing machine with a maximum load of 3000 kN was used. The stress-control loading mode was adopted with a loading rate of 0.5 MPa/s. The compressive strength was calculated by Equation (1).
(1)$$f_{\mathrm{cc}}=\frac{F}{A}$$
where *f*_cc_ is the cubic compressive strength of concrete (MPa), *F* is the applied peak load, and *A* is the sectional area of the specimen.

#### 2.4.3. Flexural Test

A flexural test was conducted according to the Chinese standard [37]. In this study, a universal testing machine with a capacity of 300 kN was used. The four-point bending tests were carried out for each specimen, and the applied load was controlled by displacement with a loading rate of 0.3 mm/min. A force transducer and three displacement meters were adopted to measure the applied load and mid-span deflection of each specimen (Figure 5). The flexural strength of each specimen can be computed by Equation (2).
(2)$$f_{f}=\frac{Fl}{bh^{2}}$$
where *f*_f_ is the flexural strength (MPa), *F* is the maximum applied load (kN), *l* is the span of the prismatic beam (300 mm), *b* is the width of the prismatic beam (100 mm), and *h* is the height of the prismatic beam (100 mm).

## 3. Results and Discussion

### 3.1. Flowability

The slump results of different mixes are shown in Figure 6. The effects of different RA replacement rates on slump values of RA concrete (RAC) are presented in Figure 6a. It can be seen that as the replacement rate of RA increased, the slump value of RAC decreased due to the relatively large water absorption and rough surface of RA. The slumps of RCFRP fiber-reinforced concrete (0% RA) and RCFRP fiber-reinforced 20% RA concrete (20% RA) with different RCFRP fiber content are shown in Figure 6b. It can be seen that with the increase of RCFRP fiber content, the slump value decreased. This is because the presence of RCFRP fibers formed a three-dimensional web [38,39,40], which hindered the flowability of concrete.

### 3.2. Compressive Strength

The compressive strengths of different mixes are illustrated in Figure 7. As shown in Figure 7a, the RA replacement rate had limited influence on the compressive strength of concrete. This is because on one hand, the high water absorption of RA decreased the water/cement ratio of concrete, which led to an increase in strength. On the other hand, RA generally exhibited lower crushing index, which reduced the compressive strength of RAC [41].

The compressive strengths of fiber-reinforced concrete and RFRAC specimens with different RCFRP fiber contents are shown in Figure 7b. It can be seen from the figure that the presence of RCFRP had a small negative influence on the compressive strength of concrete. According to the study of Mastali et al. [28], the addition of RCFRP fibers increases the porosity of concrete matrix, which leads to a reduction in compressive strength. In addition, the presence of RCFRP fibers in concrete can also constrain the lateral dilation induced by the Poisson effect when subjected to axial compression [42]. According to some experimental studies, the addition of fibers in concrete can effectively improve the compressive strength [14,16,43]. In this study, the effect of fiber addition reduced the compressive strength slightly. Further studies are required to enhance the binding between RCFRP and mortar, so as to improve the compressive strength of the RFRAC.

The failure observations of different specimens are shown in Figure 8. The mixes without RCFRP fiber, such as RA0RF0 and RA20RF0, experienced a shear failure. Surface concrete of these two mixes spalled, and the remaining specimens were in a cone shape. As for specimens with 0.5% RCFRP fibers (RA0RF5 and RA20RF5), diagonal shear cracks and concrete spalling were observed. For RA0RF10 and RA20RF10, the extent of concrete spalling was reduced, and some minor/splitting cracks could be observed. For RA0RF15 and RA20RF15, the extent of concrete spalling was moderate, and the specimens failed with several major splitting cracks. The experiments revealed that with the addition of RCFRP fibers, the extent of concrete spalling was alleviated, and the specimen was still in one piece owing to the bridging action of RCFRP fibers in concrete.

### 3.3. Flexural Performance

The flexural failure patterns of different mixes are presented in Figure 9. The mixes without RCFRP fiber (RA0RF0) experienced a brittle failure. When a flexural crack appeared near the mid-span, it propagated throughout the section, and the specimen broke into two pieces. As for the other mixes with RCFRP fiber addition (RA0RF5–RA0RF15), the failure pattern was more ductile. With the increase of applied displacement, a flexural crack also occurred near the mid-span. Nevertheless, the specimens remained with good integrity at the ultimate displacement, and the specimen could still sustain the applied load.

The size effect can influence the strength and brittleness of fiber-reinforced concrete materials. According to the work of Nguyen et al. [44], the flexural strength of fiber-reinforced concrete decreases with the increase of the size of specimens, while the brittleness increases with the increase of the size of specimens. The size effect is a very important factor that influences the performance of concrete materials. Therefore, further experimental investigation on the size effect of this material is required in future studies.

#### 3.3.1. Load–Deflection Curve

Figure 10 shows the load–deflection curves of different mixes under the four point bending test. As shown in the figure, the load–deflection curve can be divided into two stages, namely the elastic stage and the softening stage. When a relatively low load was applied, the performance of specimens with/without RCFRP fibers was similar, and all specimens exhibited elastic responses (Lines 0a and 0A in Figure 10). At this stage, there were no cracks observed in the specimens. Subsequently, with the increase of applied load, flexural cracks appeared at the bottom of the specimen. For specimens without RCFRP fiber, the flexural crack penetrated through the entire section, and the applied load dropped to 0 abruptly (Line ab in Figure 10). The failure mode of specimens without RCFRP fiber was brittle, and the failure pattern is shown in Figure 9. On the other hand, specimens with RCFRP fibers exhibited a more ductile failure mode. In the phase AB (Figure 10), the flexural crack developed to the neutral axis of the section, and the tensile force sustained by concrete gradually transferred to RCFRP fibers that stretched across the flexural crack. After point B, the crack width further developed, and the load-bearing capacity decreased more slowly than phase AB, because the RCFRP fibers could provide a relatively stable tensile force. At point C, the loading was terminated, and the specimen still had a certain residual load-bearing capacity, indicating good ductility of the specimen. The above test results show that the addition of RCFRP fiber can not only improve the flexural strength of concrete, but more importantly change the failure mode and yield a significantly better ductility.

#### 3.3.2. Flexural Strength

The effect of RA replacement rate on flexural strength is given in Figure 11a. The results show that when the RA replacement rate was 30%, the flexural strength decreased by 12.8% compared to that without RA. This is mainly because the bonding of the mortar–NA interface was better than that of the mortar–RA interface, and the tensile strength of RA was lower [45,46].

The flexural strength results of fiber-reinforced concrete and RFRAC specimens are presented in Figure 11b. It is evident that the flexural strengths of fiber-reinforced concrete and RFRAC specimens increased with the increase of RCFRP fiber content. The flexural strengths of fiber-reinforced concrete and RFRAC increased by 20.9% and 32.7%, respectively, when the RCFRP content increased from 0% to 1.5%. This is because the flexural strength of concrete was primarily determined by the tensile strength of concrete. The addition of RCFRP fiber could vastly improve the tensile strength of concrete due to the bridging effect, so that the flexural strength was significantly enhanced. It is also noteworthy that the results of flexural strength change of concrete with increase of RA were not statistical significant. This is because the flexural strength was controlled by the tensile performance of concrete, which can be influenced by various factors such as the distribution of aggregates in concrete and the interface performance between mortar and aggregate. Therefore, the flexural strength of concrete generally exhibited a relatively large deviation compared to compressive strength. Similar observation could also be found in previous studies [47].

The cross sections of specimens after the flexural test are shown in Figure 12. Figure 12b demonstrates the cross section of a specimen with 1.5% of RCFRP fiber content, where the RCFRP fibers were uniformly distributed in the cross section. Some fibers were pulled out and some were broken, indicating that CFRP fibers played a key role in constraining the development of flexural cracks.

#### 3.3.3. Flexural Stiffness

According to the load–deflection curves in Figure 9, the initial flexural stiffness varied among different specimens. To compare the initial flexural stiffness of different mixes, the secant stiffness corresponding to 0.8 times the peak flexural strength was adopted as the initial stiffness [48]. The calculated initial flexural stiffness results are shown in Figure 13.

As shown in Figure 13a, the addition of RA had a negative influence on the initial flexural stiffness of concrete. This is because the porous nature of RA and the bonding of the RA–mortar interface was weak [49], which resulted in a reduction in flexural stiffness. The initial flexural stiffness of concrete with different RCFRP content is shown in Figure 13b. It can be seen that the addition of RCFRP fibers slightly increased the stiffness of concrete.

#### 3.3.4. Flexural Toughness and Residual Load-Bearing Capacity

Flexural toughness can be used to quantify the energy dissipation capability of a material. The method according to the Chinese standard [50] was adopted to calculate the flexural toughness index of different mixes, as shown in Figure 14. The flexural toughness index is defined as the ratio of dissipated energy at the deflection of 3.0 *δ* to that when the deflection is *δ*, where *δ* is the initial crack deflection. Specifically, the flexural toughness index *I* can be calculated according to Equation (3).
(3)$$I=\frac{S_{\mathrm{OACD}}}{S_{\mathrm{OAB}}}$$
where *S*_OAB_ and *S*_OACD_ represent the enclosed areas of the load–deflection curve at the deflections of *δ* and 3.0 *δ*, respectively; *δ* is the initial crack deflection; *F*_cra_ is the initial crack strength.

The calculated flexural toughness indexes of all mixes are presented in Figure 15a. As evident in the figure, the addition of RCFRP fibers had a positive influence on the flexural toughness of concrete. For example, the flexural toughness indexes of fiber-reinforced concrete specimens with 0.5%, 1%, and 1.5% of RCFRP fiber addition were 1.23, 1.59, and 1.89 times that of the specimen without RCFRP fiber. Meanwhile, the flexural toughness indexes of RFRAC specimens with 0.5%, 1%, and 1.5% of RCFRP fiber addition were 1.32, 1.80, and 2.60 times that of the specimen without RCFRP fiber. Therefore, the addition of RCFRP fibers can largely improve the flexural toughness of concrete.

Moreover, the residual load ratio *R* was used to describe the flexural ductility of the material. As shown in Equation (4), *F*_R_ represents the residual load at the deflection of 3.0 *δ*, and *F*_MAX_ represents the maximum load–bearing capacity.
(4)$$R=\frac{F_{R}}{F_{\mathrm{MAX}}}$$

The residual load ratio of all mixes are shown in Figure 15b. It is evident that the addition of RCFRP fibers substantially improved the residual load ratio of concrete. The residual load ratio was the greatest when the volume content of RCFRP fiber was 1.5%.

The above test results show that the RFRAC concrete proposed in this paper exhibited better flexural performance than plain concrete, which makes it promising for use in civil concrete structures with high flexural performance requirements.

## 4. Conclusions

The development and mechanical properties of an eco-friendly concrete composite synthetically combined by RCFRP fibers and RA are experimentally investigated in this work. The proposed concrete composite can make extensive use of waste CFRP fibers and RA, which is beneficial to the environment. Moreover, the addition of RCFRP fibers and RA can significantly improve the flexural performance of concrete. Therefore, the eco-friendly concrete composite RFRAC is promising for use in civil infrastructures with high flexural performance requirements. Detailed conclusions can be drawn as follows:(1)The addition of RCFRP and RA decreases the flowability of concrete. Specifically, the slump value of specimen RA20RF15 (with 20% of RA replacement rate and 1.5% volume content of RCFRP fibers) is 36.0% less than that of the plain concrete specimen RA0RF0.(2)The proposed concrete composite exhibits similar compressive strength to that of plain concrete. The compressive strength of the fiber-reinforced specimens RA20RF5, RA20RF10, and RA20RF15 are 1.027, 1.005, and 0.958 times that of the reference specimen RA20RF0, respectively.(3)The addition of RA slightly decreases the flexural strength of concrete. For specimen RA20RF0, the flexural strength dropped by 6.7% compared to that of plain concrete. Nevertheless, the addition of RCFRP can largely enhance the flexural strength of concrete. For example, the flexural strength of specimen RA20RF15 increased by 23.8% compared to that of plain concrete.(4)The addition RCFRP fibers can significantly improve the flexural toughness and residual load-bearing capacity of concrete. For RFRAC specimens with 0.5%, 1%, and 1.5% of RCFRP fiber addition, the flexural toughness indexes are 1.32, 1.80, and 2.60 times that of the plain concrete specimen. Furthermore, the residual load ratio of specimen RA20RF15 is 0.69, which is significantly improved compared to the residual load ratio of 0.0 for the plain concrete specimen RA0RF0.(5)The failure patterns of mixes with/without RCFRP fibers are notably different. For the compressive test, the extent of concrete spalling of RFRAC was alleviated compared to RAC and plain concrete mixes, and the RFRAC specimens were still in one piece owing to the bridging action of RCFRP fibers in concrete. For the four-point bending test, the RFRAC specimens remained in good integrity at the ultimate displacement, and the specimen could still sustain the applied load, while the RAC and plain concrete mixes broke into two pieces when a flexural crack occurred.

## Figures and Tables

**Figure 1 materials-13-04592-f001:**
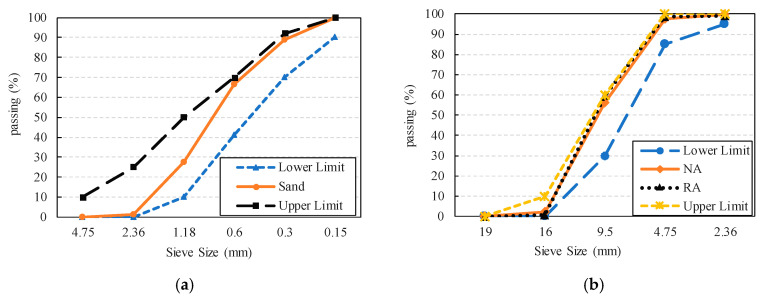
Particle-size distributions: (**a**) particle-size distribution of sand; (**b**) particle-size distribution of natural aggregate (NA) and recycled aggregate (RA).

**Figure 2 materials-13-04592-f002:**
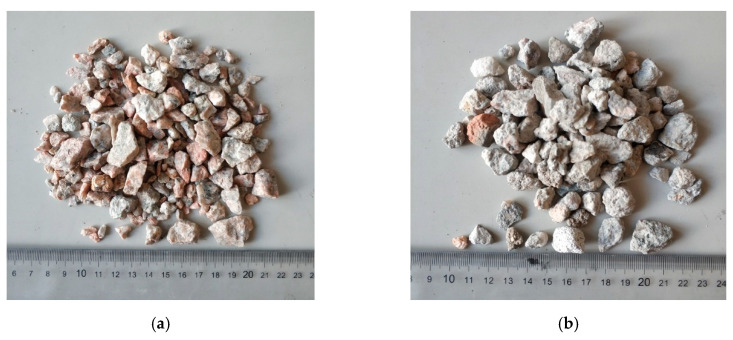
Material and machine: (**a**) NA; (**b**) RA; (**c**) waste road concrete; (**d**) crushing machine.

**Figure 3 materials-13-04592-f003:**
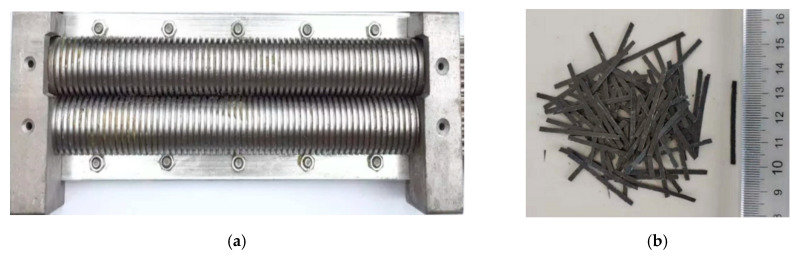
Mechanical recycling of carbon fiber reinforced polymer CFRP fibers: (**a**) shredding tool; (**b**) short cut recycled CFRP (RCFRP) fibers.

**Figure 4 materials-13-04592-f004:**
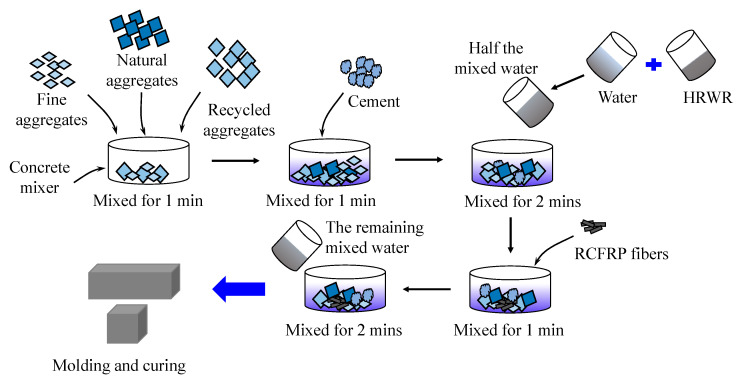
Concrete preparation procedure.

**Figure 5 materials-13-04592-f005:**
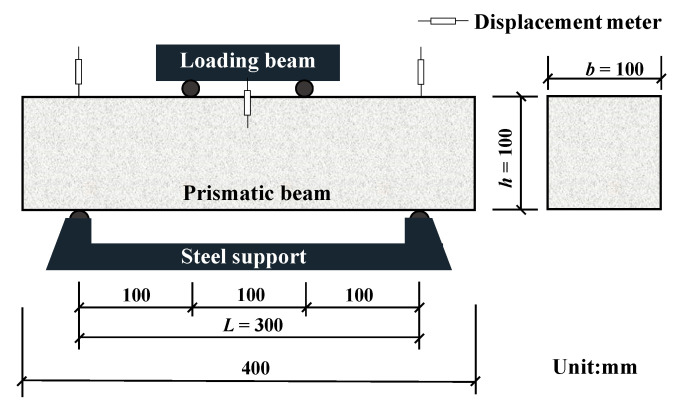
Setup of the flexural test.

**Figure 6 materials-13-04592-f006:**
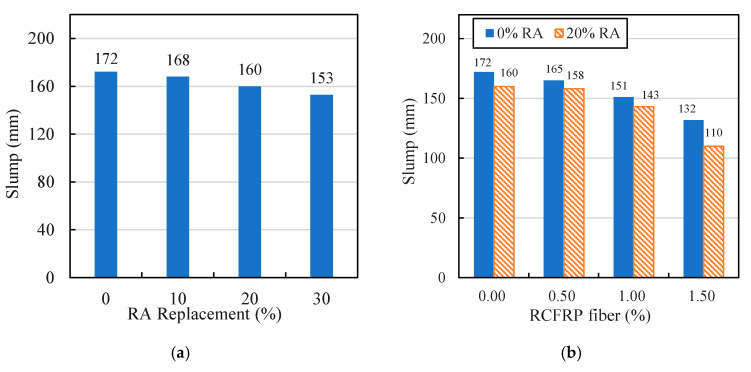
Slump results: (**a**) RAC; (**b**) fiber-reinforced concrete and RFRAC.

**Figure 7 materials-13-04592-f007:**
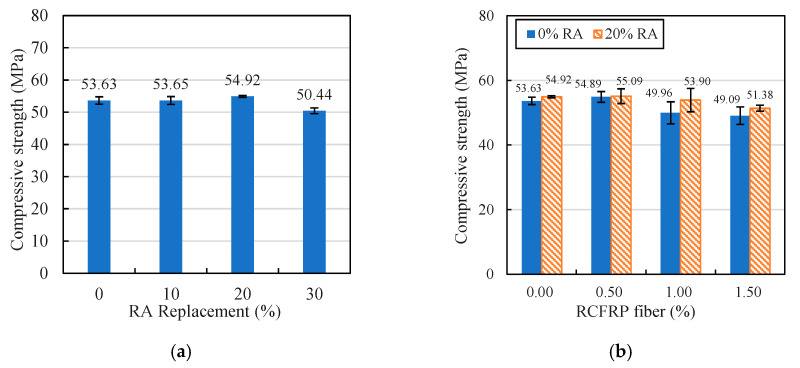
Compressive strength: (**a**) RAC; (**b**) fiber-reinforced concrete and RFRAC.

**Figure 8 materials-13-04592-f008:**
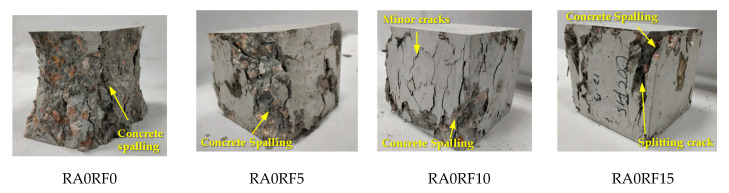
Failure pattern of different specimens.

**Figure 9 materials-13-04592-f009:**
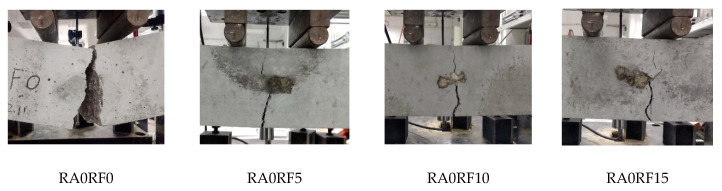
The flexural failure patterns of different specimens.

**Figure 10 materials-13-04592-f010:**
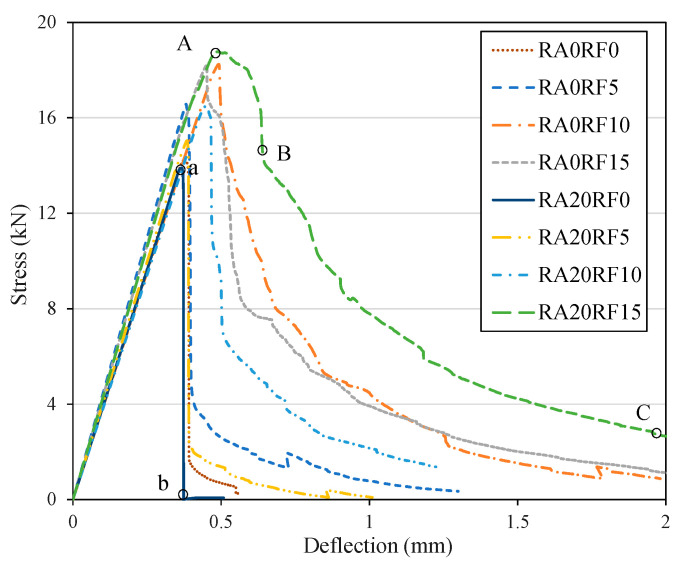
Load–deflection curves.

**Figure 11 materials-13-04592-f011:**
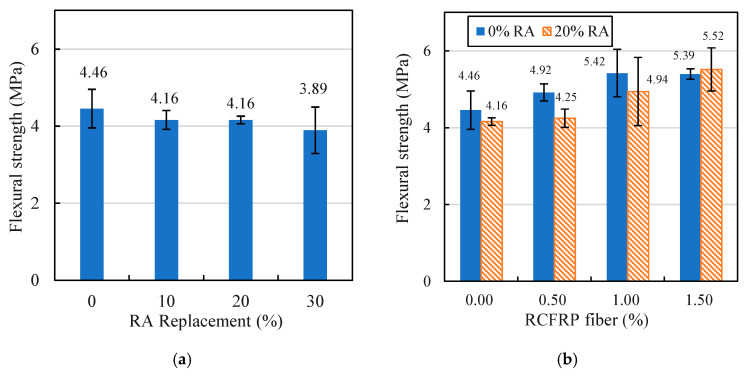
Flexural strengths of different specimens: (**a**) RAC; (**b**) fiber-reinforced concrete and RFRAC.

**Figure 12 materials-13-04592-f012:**
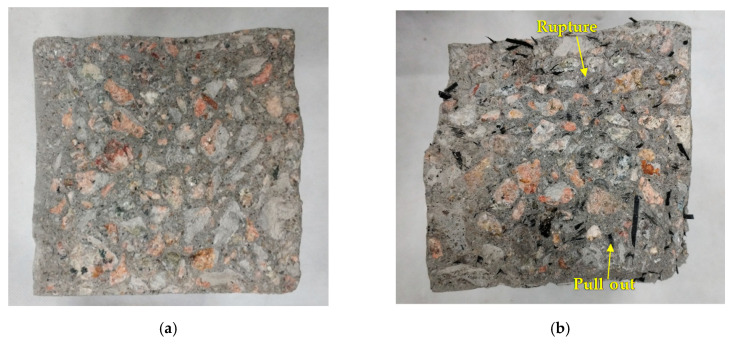
Fracture surfaces: (**a**) RA0RF0; (**b**) RA0RF15.

**Figure 13 materials-13-04592-f013:**
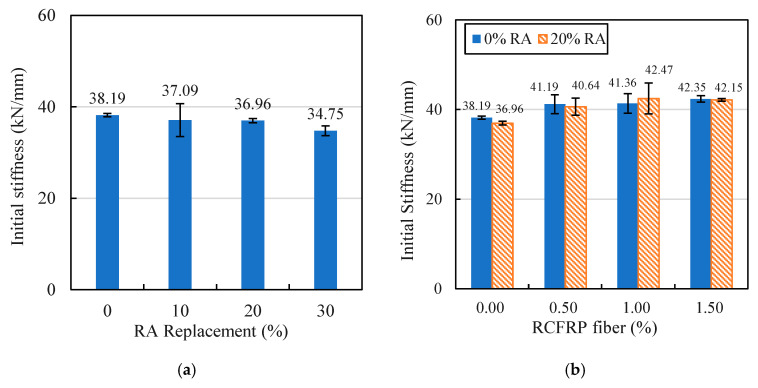
Comparison of flexural stiffness results: (**a**) RAC; (**b**) fiber-reinforced concrete and RFRAC.

**Figure 14 materials-13-04592-f014:**
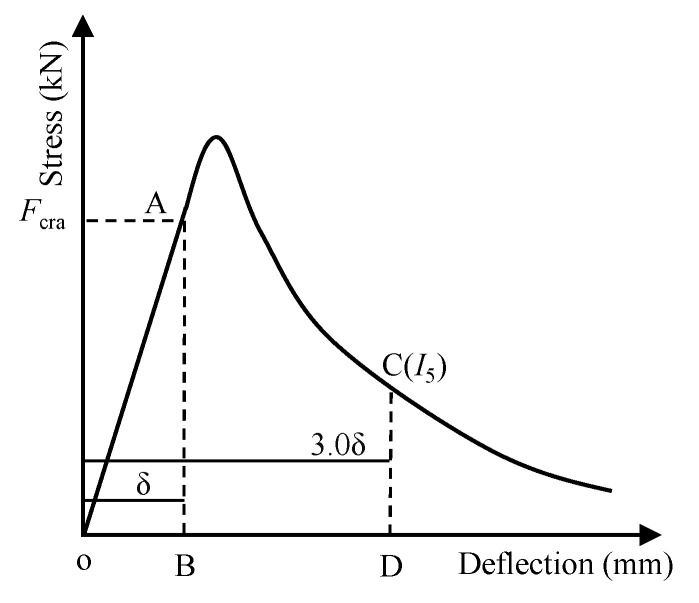
Calculation of flexural toughness index.

**Figure 15 materials-13-04592-f015:**
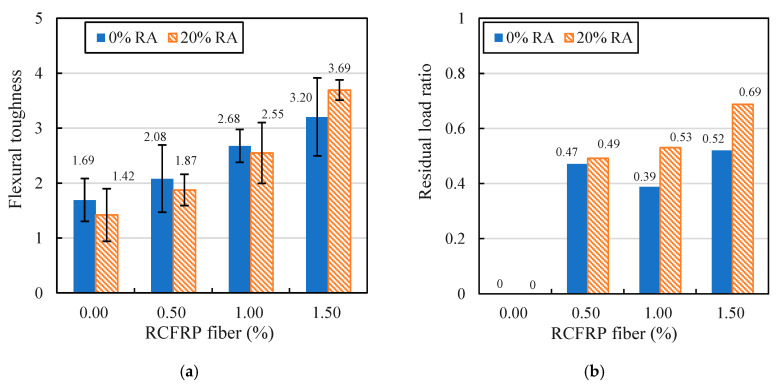
Comparison of flexural indexes and residual load ratios: (**a**) flexural toughness index; (**b**) residual load ratio.

**Table 1 materials-13-04592-t001:** Chemical composition of ordinary Portland cement (OPC).

Ingredient	CaO	SiO_2_	Al2O_3_	Fe_2_O_3_	MgO	SO_3_	K_2_O	Na_2_O	LOI
Content (mass %)	64.42	20.52	5.62	3.78	2.11	2.10	0.28	0.20	0.87

**Table 2 materials-13-04592-t002:** Properties of NA and RA.

Material Properties	NA	RA
Water absorption (%)	1.44	6.02
Apparent density (kg/m^3^)	2617	2639
Crushing index (%)	11.36	19.49

**Table 3 materials-13-04592-t003:** The mixture designs of different concrete samples.

Sample ID	RCFRP (kg/m^3^)	NA (kg/m^3^)	RA (kg/m^3^)	Sand (kg/m^3^)	Water (kg/m^3^)	Cement (kg/m^3^)	Plasticizer (kg/m^3^)
RA0RF0	N/A	1078	N/A	590	215	540	1.05
RA10RF0	N/A	970.2	108.7	590	215	540	1.05
RA20RF0	N/A	862.4	217.4	590	215	540	1.05
RA30RF0	N/A	754.6	326.1	590	215	540	1.05
RA0RF5	8.5	1072.64	N/A	587.06	213.93	537.31	1.04
RA0RF10	17.0	1067.33	N/A	584.16	212.87	534.65	1.04
RA0RF15	25.5	1062.07	N/A	581.28	211.82	532.02	1.03
RA20RF5	8.5	858.11	216.32	587.06	213.93	537.31	1.04
RA20RF10	17.0	853.86	215.25	584.16	212.87	534.65	1.04
RA20RF15	25.5	849.66	214.19	581.28	211.82	532.02	1.03

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
