# Peer review of "Study of Mechanical Properties of an Eco-Friendly Concrete Containing Recycled Carbon Fiber Reinforced Polymer and Recycled Aggregate"

_materials, 2020, doi:10.3390/ma13204592_

Round 1
Reviewer 1 Report
Reviewed manuscript titled "Development and Mechanical Properties of an Eco-friendly Concrete Containing Recycled Carbon Fiber Reinforced Polymer and Recycled Aggregate" perfectly fit to Journal „Materials”.
Introduction is very well prepared. Authors present interesting conception use both Recycled Carbon Fiber and Recycled Aggregate to improve strength and earthquake behavior. Aspect of earthquake behavior should be much more highlighted. I am not see using of this material in important infrastructure structures, like bridges and pavements (due to low durability of RA concrete). I have seen above suggestion also in line 290-292.
I concern about correctness of concluding and discussion of results, because design of concrete with additives of RCFRP do not consider additional volume of fiber in designed concrete. Hence wrong interpretation of result in line 177. Very useful confirmation of proportion ingredients change will be presenting a density measurements of concrete mixture.
Results of investigation are well presented and discussed compared to other publications, but results of flexural strength change of concrete with increase of RA is not statistical significant.
Conclusion are based on measurements results. I suggest more precise describe conclusion 2 (line 305-306). Additionally I have not seen optimization evidence to conclusion 5 (is known, that, more fibers - better properties, why not 2% of fiber?)
I suggest minor revision of manuscript.
Author Response
Point 1: Reviewed manuscript titled "Development and Mechanical Properties of an Eco-friendly Concrete Containing Recycled Carbon Fiber Reinforced Polymer and Recycled Aggregate" perfectly fit to Journal “Materials”.
Response 1: The authors appreciate the positive comment of the reviewer. The manuscript has been improved according to the reviewer’s comments.
Point 2: Introduction is very well prepared. Authors present interesting conception use both Recycled Carbon Fiber and Recycled Aggregate to improve strength and earthquake behavior. Aspect of earthquake behavior should be much more highlighted. I am not see using of this material in important infrastructure structures, like bridges and pavements (due to low durability of RA concrete). I have seen above suggestion also in line 290–292.
Response 2: The authors appreciate the constructive comment of the reviewer. The addition of fibers in concrete can improve the seismic performance of concrete structures. For example, Xu et al. [15] investigated the seismic performance of concrete columns with the addition of steel fibers. The results show that the existence of steel fiber could prevent the cracked concrete from spalling and delay the bulking of longitudinal reinforcement. Huang et al. [16] studied the seismic performance of concrete columns with steel–polypropylene hybrid fibers. The columns exhibited a notable synergetic effect in terms of ductility and energy dissipation capacity, particularly for columns with a higher axial compression ratio. The improvement of seismic performance are primarily attributes to the bridging action provided by fibers [17]. It is believed that other fibers, such as carbon fiber, can also be adopted to improve the seismic performance of concrete structures. The above description can be found in Lines 47 to 57.
As suggested by the reviewer, the addition of RA can influence the durability of concrete [11]. The actual application of a new material requires comprehensive investigations on the durability, reliability, and affordability of the material, which is out the scope of this study. Considering that flexural performance of the material is improved compared with plain concrete, the proposed material is expected to have possible application in civil concrete structures with high flexural performance requirement. The above manuscript has been revised in Lines 21 to 23; Lines 338 to 340 and Lines 346 to 348.
- Kou, S.C.; Poon, C.S. Enhancing the durability properties of concrete prepared with coarse recycled aggregate. Constr. Build. Mater. 2012, 35, 69–76. doi:10.1016/j.conbuildmat.2012.02.032.
- Xu, S.; Wu, C.; Liu, Z.; Han, K.; Su, Y.; Zhao, J.; Li, J. Experimental investigation of seismic behavior of ultra-high performance steel fiber reinforced concrete columns. Eng. Struct. 2017, 152, 129–148. doi:10.1016/j.engstruct.2017.09.007.
- Huang, L.; Xu, L.; Chi, Y.; Xu, H. Experimental investigation on the seismic performance of steel–polypropylene hybrid fiber reinforced concrete columns. Constr. Build. Mater. 2015, 87, 16–27, doi:10.1016/j.conbuildmat.2015.03.073.
- Li, V.C.; Stang, H.; Krenchel, H. Micromechanics of crack bridging in fibre-reinforced concrete. Mater. Struct. 1993, 26(8), 486–494, doi:10.1007/BF02472808.
Point 3: I concern about correctness of concluding and discussion of results, because design of concrete with additives of RCFRP do not consider additional volume of fiber in designed concrete. Hence wrong interpretation of result in line 177. Very useful confirmation of proportion ingredients change will be presenting a density measurements of concrete mixture.
Response 3: The authors appreciate the reviewer’s comment. Additional volume of fiber has been considered in the designed fiber-reinforced concrete. In Table 3 of the original manuscript, to demonstration the differences of mixes with/without fibers, the amount of ingredients other than fibers and aggregates remains the same. In fact, the volume of a mix with fibers is slightly larger than that without fiber. For example, the volume of RA0RF10 is 0.65% larger than that of RA0RF0 when considering the additional fiber volume and the slight density change of specimens. Therefore, according to the reviewer’s comment, the quantity of each ingredient for unit volume of concrete is updated considering the addition of fiber volume and density change in Table 3.
For the interpretation of experiment results, several studies also indicate that the presence of RCFRP fibers can hinder the flowability of concrete by forming the three-dimensional web [35-37], which supports the interpretation of result in Line 177 of the original manuscript. The above description can be found in Lines 215 to 218 of the revised manuscript.
The addition of RCFRP fibers has limited influence on the density of the fiber-reinforced concrete. For example, the density of RA0RF10 is 2385 kg/m3, which is similar to the plain concrete specimen RA0RF0 (2400.5 kg/m3). The above description can be found in Lines 167 to 169.
- Aslani, F.; Kelin, J. Assessment and development of high-performance fibre-reinforced lightweight self-compacting concrete including recycled crumb rubber aggregates exposed to elevated temperatures. J. Clean. Prod. 2018, 200, 1009–1025. doi:10.1016/j.jclepro.2018.07.323.
- Aydin, A.C. Self compactability of high volume hybrid fiber reinforced concrete. Constr. Build. Mater. 2007, 21(6), 1149–11545. doi:10.1016/j.conbuildmat.2006.11.017.
- Chen, B.; Liu, J. Contribution of hybrid fibers on the properties of the high-strength lightweight concrete having good workability. Cem. Concr. Res. 2020, 35(5), 913–917. doi:10.3390/ma13040868.
Table 3. The mixture designs of different concrete samples
|
Sample ID |
RCFRP (kg/m3) |
NA (kg/m3) |
RA (kg/m3) |
Sand (kg/m3) |
Water (kg/m3) |
Cement (kg/m3) |
Plasticizer (kg/m3) |
|
RA0RF0 |
N/A |
1078.00 |
N/A |
590.00 |
215.00 |
540.00 |
1.05 |
|
RA10RF0 |
N/A |
970.20 |
108.70 |
590.00 |
215.00 |
540.00 |
1.05 |
|
RA20RF0 |
N/A |
862.40 |
217.40 |
590.00 |
215.00 |
540.00 |
1.05 |
|
RA30RF0 |
N/A |
754.60 |
326.10 |
590.00 |
215.00 |
540.00 |
1.05 |
|
RA0RF5 |
8.5 |
1072.64 |
N/A |
587.06 |
213.93 |
537.31 |
1.04 |
|
RA0RF10 |
17.0 |
1067.33 |
N/A |
584.16 |
212.87 |
534.65 |
1.04 |
|
RA0RF15 |
25.5 |
1062.07 |
N/A |
581.28 |
211.82 |
532.02 |
1.03 |
|
RA20RF5 |
8.5 |
858.11 |
216.32 |
587.06 |
213.93 |
537.31 |
1.04 |
|
RA20RF10 |
17.0 |
853.86 |
215.25 |
584.16 |
212.87 |
534.65 |
1.04 |
|
RA20RF15 |
25.5 |
849.66 |
214.19 |
581.28 |
211.82 |
532.02 |
1.03 |
Point 4: Results of investigation are well presented and discussed compared to other publications, but results of flexural strength change of concrete with increase of RA is not statistical significant.
Response 4: The authors appreciate the reviewer’s comment. As suggested by the reviewer, the results of flexural strength change of concrete with increase of RA is not statistical significant. This is because the flexural strength is controlled by the tensile performance of concrete, which can be influenced by various factors such as the distribution of aggregates in concrete and the interface performance between mortar and aggregate. Therefore, the flexural or tensile strengths of concrete generally exhibits a relatively large deviation compared to compressive strength. Similar observation can also be found in previous studies [43]. The above discussion can be found in Lines 285 to 291.
- Bai, G.; Zhu, C.; Liu, C.; Liu, B. An evaluation of the recycled aggregate characteristics and the recycled aggregate concrete mechanical properties. Constr. Build. Mater. 2020, 240, 117978. doi:10.1016/j.conbuildmat.2019.117978.
Point 5: Conclusion are based on measurements results. I suggest more precise describe conclusion 2 (line 305–306). Additionally, I have not seen optimization evidence to conclusion 5 (is known, that, more fibers - better properties, why not 2% of fiber?)
Response 5: The authors appreciate the reviewer’s constructive comments. More precise description of Conclusion 2 is presented as follows:
The proposed concrete composite exhibits similar compressive strength to that of plain concrete. The compressive strength of the fiber-reinforced specimens RA20RF5, RA20RF10, and RA20RF15 are 1.027, 1.005, and 0.958 times that of the reference specimen RA20RF0, respectively. The above description can be found in Lines 352 to 355.
This study investigated the mechanical performance of RA concrete with the RCFRP fiber addition of 0.5% – 1.5%. In this range, the concrete composite exhibits better performance with the increase of fiber addition. As suggested by the reviewer, in order to obtain to the optimal mix, further experiment is required. Therefore, the conclusion 5 is removed.

Reviewer 2 Report
The manuscript covers an extensive laboratory investigation of mechanical properties of an eco-friendly concrete containing Recycled Carbon Fiber Reinforced Polymer and Recycled Aggregate. The research deals an interesting and relevant aspect of recycling in order to achieve economic and environmental sustainability of the construction industry.
However, key deficiencies especially in the mix design have been identified in the research which need to be addressed. Find below for your consideration my comments with request for clarifications and/ or further improvement of the manuscript:
- I suggest that the title should not have both “development” and “mechanical properties” since any development of special concrete would involve assessing mechanical properties. It can either be “Study of Mechanical Properties of an Eco-friendly Concrete..........” or “Development of an Eco-friendly Concrete..........”. The same applies to conclusion in line 294.
- In line 41-42, clearly point out that flexural and splitting strengths are for “RA” concrete.
- In line 50-52, provide the current state of CFRP waste products as well, in order to supplement the projections being given. Also mention the exact environmental impact of waste CFRP.
- In line 63, provide the range of lengths for the short fibres discussed.
- In line 74-75, it is concluded that the mechanical properties of RA concrete are lower than those of conventional concrete. However, no literature has been provided to back this statement and almost all literature provided is focused on fibre reinforced concrete. In fact, only reference [9] has been provided to indicate that RA concrete generally exhibits relatively poor mechanical properties. Provide more literature, discussion and references to support your statement.
- According to line 102-103, RA used were dried and therefore in an oven dry (OD) moisture state. Also as noted line 105-106, the RA had a high water absorption which was 4.18 times that of NA. How then in the mix design was this high water absorption of RA compensated for to avoid the mixing water being absorbed by RA which would then affect the W/C ratio used and therefore strength of concrete produced?
- In line 123-125, how was 0% and 20% RA replacement rate selected to study the effect of RCFRP fibre contents? What criteria guided the decision for eliminating 10% and 30% RA replacement rate?
- In line 127-128, what amount of polycarboxylate-based high range water reducer (HRWR) was applied (as percentage of cement content) in the mix? Plasticizers are usually expressed as a percentage of cement content.
- Figures 6b, 7b, 11b and 13b are for what % of RA replacement rate? Is it 0% or 20% RA replacement rate?
- In line 182-183, it is clearly pointed out the “the high water absorption of RA decreases the water-cement ratio of concrete, which leads to an increase in strength”. This further brings out my concern about the mix design which I already highlighted in comment 6 above. It simply implies that your mixes effectively had different W/C ratios as a result of RA absorbing the mixing water. So, in other words, mixes that had higher RA percentage replacements had lower W/C ratios since more mixing water was absorbed by RA. It is therefore not clear how you will compare mechanical properties of different mixes each with a varying W/C ratio from the other. For comparison purposes, it is expected that mixes should have the same boundary conditions, and thus the W/C ratio should be maintained constant for all mixes in your study. This is usually achieved in laboratory mixes by having the aggregates in saturated surface dry (SSD) moisture state to ensure that they will neither absorb not release water to the mix. However, the absorption of aggregates changes the W/C ratio for the different mixes. Please provide an explanation about this in your mix design.
- In line 191-192, constraining of cracking on concrete is attributed to improvement of compressive strength. However, cracking is more related to the tensile strength of concrete than compressive strength.
- In line 198, it is stated that “the failure mode changed significantly”. However, the mode of failure is the same but what is different is the rate/level of failure which is more pronounced in plain than fibre reinforced concrete.
- In line 299-300, I think it better to only have a general inference to RFRAC being promising for use in civil concrete structures without giving specific examples (earthquake-resistant structures, concrete pavements, and impact barriers) since no discussion has been given earlier in the manuscript in relation specific examples. Therefore, this inference should not now be introduced for the first time in the conclusion. The same applies for line 24 in the abstract.
Author Response
Point 1: The manuscript covers an extensive laboratory investigation of mechanical properties of an eco-friendly concrete containing Recycled Carbon Fiber Reinforced Polymer and Recycled Aggregate. The research deals an interesting and relevant aspect of recycling in order to achieve economic and environmental sustainability of the construction industry.
However, key deficiencies especially in the mix design have been identified in the research which need to be addressed. Find below for your consideration my comments with request for clarifications and/ or further improvement of the manuscript:
Response 1: The authors appreciate the constructive comments of the reviewer. The manuscript has been improved according to the reviewer’s suggestions.
Point 2: I suggest that the title should not have both “development” and “mechanical properties” since any development of special concrete would involve assessing mechanical properties. It can either be “Study of Mechanical Properties of an Eco-friendly Concrete..........” or “Development of an Eco-friendly Concrete..........”. The same applies to conclusion in line 294.
Response 2: According to the suggestion of the reviewer, the title has been modified to “Study of Mechanical Properties of an Eco-friendly Concrete Containing Recycled Carbon Fiber Reinforced Polymer and Recycled Aggregate”.
Point 3: In line 41–42, clearly point out that flexural and splitting strengths are for “RA” concrete. The research of Akça et al. indicates that polypropylene fibers can improve the flexural and splitting strengths of recycle concrete [11].
Response 3: The authors appreciate the comment of the reviewer. In the research of Akça et al., flexural and splitting strengths are for “RA” concrete. The revised sentence is “the research of Akça et al. indicates that polypropylene fibers can improve the flexural and splitting strengths of RA concrete [13]”. The revision can be found in Lines 44 to 45.
- Akça, K.R.; Çakır, Ö.; İpek, M. Properties of polypropylene fiber reinforced concrete using recycled aggregates. Constr. Build. Mater. 2015, 98, 620–630, doi:10.1016/j.conbuildmat.2015.08.133.
Point 4: In line 50–52, provide the current state of CFRP waste products as well, in order to supplement the projections being given. Also mention the exact environmental impact of waste CFRP.
Response 4: The authors appreciate the comment of the reviewer. According to the work of Meng et al [19], an estimated 62,000 tonnes of CFRP in production and end-of-life waste will be generated by 2020.
Traditional disposal methods for carbon fibers include waste landfill or combustion. Nevertheless, these disposal solutions can pose a significant impact on the environment, such as land use, groundwater pollution or harmful gas emissions [22].
The above description can be found in Lines 62 to 64 and Lines 66 to 68.
- Meng, F.; Olivetti, E. A.; Zhao, Y.; Chang, J.C.; Pickering, S.J.; McKechnie, J. Comparing life cycle energy and global warming potential of carbon fiber composite recycling technologies and waste management options. ACS Sustain. Chem. Eng. 2018, 6(8), 9854–9865. doi:10.1021/acssuschemeng.8b01026.
- Akbar, A.; Liew, K.M. Assessing recycling potential of carbon fiber reinforced plastic waste in production of eco-efficient cement-based materials. J. Clean. Prod. 2020, 274, 123001. doi:10.1016/j.jclepro.2020.123001.
Point 5: In line 63, provide the range of lengths for the short fibres discussed.
Response 5: The authors appreciate the comment of the reviewer. The length of short fibers obtained through mechanical recycling can be affected by the recycle machinery and the size of waste CFRP raw materials. The research of Gopalraj et al. [25] shows that the maximum length of mechanically recycled fibers is 100 mm, and the study of Mastali et al. [26] adopts short fibers with the length of 30 mm. In this paper, recycled short fibers with the length of 35 mm are adopted. Therefore, the length of mechanically recycled short fibers usually ranges from 0 – 100 mm. The above description can be found in Lines 80 to 86.
- Gopalraj, S.K.; Kärki, T. A review on the recycling of waste carbon fibre/glass fibre-reinforced composites: fibre recovery, properties and life-cycle analysis. SN Appl. Sci. 2020, 2(3), 1–21. doi:10.1016/j.supflu.2019.104607.
- Mastali, M.; Dalvand, A.; Sattarifard, A. The impact resistance and mechanical properties of the reinforced self-compacting concrete incorporating recycled CFRP fiber with different lengths and dosages. Compos. Part B Eng. 2017, 112, 74–92, doi:10.1016/j.compositesb.2016.12.029.
Point 6: In line 74–75, it is concluded that the mechanical properties of RA concrete are lower than those of conventional concrete. However, no literature has been provided to back this statement and almost all literature provided is focused on fibre reinforced concrete. In fact, only reference [9] has been provided to indicate that RA concrete generally exhibits relatively poor mechanical properties. Provide more literature, discussion and references to support your statement.
Response 6: The authors appreciate the comment of the reviewer. The work of Evangelista et al. [10] indicates that with the increases of RA replacement, the tensile splitting strength and modulus of elasticity decrease. The study of Kou et al. [11] shown that the drying shrinkage and creep of concrete increase with the increase of RA content, while the compressive strength and durability decrease with the increase of RA content. The above discussion has been added in Lines 37 to 41 of the revised manuscript.
- Evangelista, L.; de Brito, J. Mechanical behaviour of concrete made with fine recycled concrete aggregates. Cem. Concr. Compos. 2007, 29(5), 397–401. doi:10.1016/j.cemconcomp.2006.12.004.
- Kou, S.C.; Poon, C.S. Enhancing the durability properties of concrete prepared with coarse recycled aggregate. Constr. Build. Mater. 2012, 35, 69–76. doi:10.1016/j.conbuildmat.2012.02.032.
Point 7: According to line 102–103, RA used were dried and therefore in an oven dry (OD) moisture state. Also as noted line 105–106, the RA had a high water absorption which was 4.18 times that of NA. How then in the mix design was this high water absorption of RA compensated for to avoid the mixing water being absorbed by RA which would then affect the W/C ratio used and therefore strength of concrete produced?
Response 7: The authors appreciate the constructive comment of the reviewer. When designing the concrete mixes, the authors have considered two ideas. The first idea is to keep the amount of RCFRP fibers and RA as the control variables and the amount of other ingredients stays constant. For this idea, the absolute water content for all mixes are the same. The second idea is similar to the one suggested by the reviewer, where the actual w/c ratio is kept constant for all mixes. For this idea, the absolute water content of different mixes is different considering the water absorption of other materials. Note that both RA and RCFRP fibers can absorb water and the actual w/c ratio is difficult to control. The authors finally select the first idea in designing the concrete mixes. The above discussion has been added in Lines 145 to 152 of the revised manuscript.
Point 8: In line 123–125, how was 0% and 20% RA replacement rate selected to study the effect of RCFRP fibre contents? What criteria guided the decision for eliminating 10% and 30% RA replacement rate?
Response 8: The authors appreciate the comment of the reviewer. For 10% replacement rate, the amount of RA used is limited, which cannot facilitates the large-scale use RA in construction. On the other hand, RA has the characteristics of low density, low wear resistance, compressive strength, high absorption and porosity, [32], too much addition of RA can affect the durability, mechanical properties and stability of RAC. According to the research of Tam et al. [2], up to 20% of RA replacement rate can be used for most applications including structural. Therefore, the mix with RA replacement rate of 20% is selected to further investigate the synergetic effect with RCFRP fibers. The above discussion has been added in Lines 156 to 162 of the revised manuscript.
- Tam, V.W.Y.; Soomro, M.; Evangelista, A.C.J. A review of recycled aggregate in concrete applications (2000–2017). Constr. Build. Mater. 2018, 172, 272–292, doi:10.1016/j.conbuildmat.2018.03.240.
- Zaetang, Y.; Sata, V.; Wongsa, A.; Chindaprasirt, P. Properties of pervious concrete containing recycled concrete block aggregate and recycled concrete aggregate. Constr. Build. Mater. 2016, 111, 15–21. doi:10.1016/j.conbuildmat.2016.02.060.
Point 9: In line 127–128, what amount of polycarboxylate-based high range water reducer (HRWR) was applied (as percentage of cement content) in the mix? Plasticizers are usually expressed as a percentage of cement content.
Response 9: The authors appreciate the comment of the reviewer. According to the suggestion of the reviewer, the mass ratio of HRWR to cement is 0.194%. The revision can be found in Lines 164 to 166 of the revised manuscript.
Point 10: Figures 6b, 7b, 11b and 13b are for what % of RA replacement rate? Is it 0% or 20% RA replacement rate?
Response 10: The authors appreciate the constructive comment of the reviewer. To make the legend self-evident, the mixes with no RA addition are labelled “0% RA”, and the mixes with 20% RA addition are labelled “20% RA”. The revised figures can be found in Lines 219, 226, 276 and 303.
Point 11: In line 182–183, it is clearly pointed out the “the high water absorption of RA decreases the water-cement ratio of concrete, which leads to an increase in strength”. This further brings out my concern about the mix design which I already highlighted in comment 6 above. It simply implies that your mixes effectively had different W/C ratios as a result of RA absorbing the mixing water. So, in other words, mixes that had higher RA percentage replacements had lower W/C ratios since more mixing water was absorbed by RA. It is therefore not clear how you will compare mechanical properties of different mixes each with a varying W/C ratio from the other. For comparison purposes, it is expected that mixes should have the same boundary conditions, and thus the W/C ratio should be maintained constant for all mixes in your study. This is usually achieved in laboratory mixes by having the aggregates in saturated surface dry (SSD) moisture state to ensure that they will neither absorb not release water to the mix. However, the absorption of aggregates changes the W/C ratio for the different mixes. Please provide an explanation about this in your mix design.
Response 11: The authors appreciate the constructive comment of the reviewer. When designing the concrete mixes, the authors have considered two ideas. The first idea is to keep the amount of RCFRP fibers and RA as the control variables and the amount of other ingredients stay constant. For this idea, the absolute water content for all mixes are the same. The second idea is similar to the one suggested by the reviewer, where the actual w/c ratio is kept constant for all mixes. For this idea, the absolute water content of different mixes is different considering the water absorption of other materials. Note that both RA and RCFRP fibers can absorb water and the actual w/c ratio is difficult to control. The authors finally select the first idea in designing the concrete mixes. The above discussion has been added in Lines 145 to 152 of the revised manuscript.
Point 12: In line 191–192, constraining of cracking on concrete is attributed to improvement of compressive strength. However, cracking is more related to the tensile strength of concrete than compressive strength.
Response 12: The authors appreciate the constructive comment of the reviewer. The authors agree that the RCFRP fibers in concrete can constrain the development of cracks, which helps to improve the tensile and flexural strengths. Moreover, the presence of fibers in concrete can also constrain the lateral dilation induced by Poisson effect when subjected to axial compression [39]. According to some experimental studies, the addition of fibers in concrete can effectively improve the compressive strength of concrete [12,14,40]. Considering that the original statement is misleading, the above description has been added to the revised manuscript in Lines 231 to 235.
- Carneiro, J.A.; Lima, P.R.L.; Leite, M.B.; Toledo Filho, R.D. Compressive stress–strain behavior of steel fiber reinforced-recycled aggregate concrete. Cem. Concr. Compos. 2014, 46, 65–72, doi:0.1016/j.cemconcomp.2013.11.006.
- Prasad, M.L.V.; Rathish Kumar, P. Strength studies on glass fiber reinforced recycled aggregate concrete. Asian. J. Civ. Eng. (Build. Hous.) 2007, 8, 677–690.
- Han B.; Xiang, T.Y. Axial compressive stress-strain relation and Poisson effect of structural lightweight aggregate concrete. Constr. Build. Mater. 2017, 146, 338–343. doi:10.1016/j.conbuildmat.2017.04.101.
- Xiong, C.; Li, Q.; Lan, T.; Li, H.; Long, W.; Xing, F. Sustainable use of recycled carbon fiber reinforced polymer and crumb rubber in concrete: mechanical properties and ecological evaluation. J. Clean. Prod. 2021, 279, 123624. doi:10.1016/j.jclepro.2020.123624.
Point 13: In line 198, it is stated that “the failure rate/level changed significantly”. However, the mode of failure is the same but what is different is the rate/level of failure which is more pronounced in plain than fibre reinforced concrete.
Response 13: The authors appreciate the comment of the reviewer. As suggested by the reviewer, the specimens still suffered compressive failure mode, while exhibits different failure levels. The plain concrete specimen (R0CF0) experienced severe compressive failure with extensive concrete spalling, while the fiber-reinforced concrete specimen is still in one piece due to the bridging action of RCFRP fibers in concrete. The revised description can be found in Lines 238 to 242.
Point 14: In line 299–300, I think it better to only have a general inference to RFRAC being promising for use in civil concrete structures without giving specific examples (earthquake-resistant structures, concrete pavements, and impact barriers) since no discussion has been given earlier in the manuscript in relation specific examples. Therefore, this inference should not now be introduced for the first time in the conclusion. The same applies for line 24 in the abstract.
Response 14: The authors appreciate the constructive comment of the reviewer. This study primarily investigates the mechanical properties of the concrete material. The results reveal that flexural performance of the material is improved compared with plain concrete. Therefore, the proposed material is expected to have possible application in civil concrete structures with high flexural performance requirement. The above manuscript has been revised in Lines 21 to 23; Lines 338 to 340 and Lines 346 to 348.
Reviewer 3 Report
Nice topic of high relevance and interest to readers
Revise end of abstract to be more quantitative
I have no other substantive comments on the text
Fig. 2 images need to be of better quality
Fig. 3b image needs to be of better quality
Fig. 8 images RA20RF0 and RA0RF0 need to be of better quality
Fig. 10 images need to be of better quality
Fig. 12b change the labeling as the red text directly on the image is unreadable
Always check the pdf when generated from the system as this one has separated headings from the text (e.g. line 192, line 217 & 218)
Line 272 add comma before “because”
Author Response
1.Revise end of abstract to be more quantitative.
The authors appreciate the constructive comment of the reviewer. The revised abstract is as follow:
This study investigates the feasibility of collaborative use of recycled carbon fiber reinforced polymer (RCFRP) fibers and recycled aggregate (RA) in concrete, which is called RCFRP fiber reinforced RA concrete (RFRAC). The mechanical properties of the composite are studied through experimental investigation, considering different RCFRP fiber contents (0%, 0.5%, 1.0%, and 1.5% by volume) and different RA replacement rates (0%, 10%, 20% and 30% by volume). Specifically, ten different mixes are designed to explore the flowability, compressive and flexural strengths of the proposed composite. Experimental results indicate that the addition of RCFRP fibers and RA has a relatively small influence on the compressive strength of concrete (less than 5%). Moreover, the addition of RA slightly decreases the flexural strength of concrete, while the addition of RCFRP fibers can significantly improve the flexural performance. For example, the flexural strength of RA concrete with 1.5% RCFRP fiber addition increased by 32.7%. Considering the good flexural properties of the composite and its potential in reducing waste CFRP and construction solid waste, the proposed RFRAC is promising for use in civil concrete structures with high flexural performance requirement.
2.Fig. 2 images need to be of better quality; Fig. 3b image needs to be of better quality; Fig. 8 images RA20RF0 and RA0RF0 need to be of better quality; Fig. 10 images need to be of better quality; Fig. 12b change the labeling as the red text directly on the image is unreadable.
The authors appreciate the constructive comment of the reviewer. The above mentioned images have been replaced with high resolution ones and the labelling in Fig. 12b has been changed. The above modifications can be found in Pages 4, 5, 9, 10, and 12 of the revised manuscript.
3.Always check the pdf when generated from the system as this one has separated headings from the text (e.g. line 192, line 217 & 218).
The authors appreciate the constructive comment of the reviewer. The formatting of the manuscript has been carefully checked and revised according to the suggestion of the reviewer.
4.Line 272 add comma before “because”.
The authors appreciate the comment of the reviewer. The manuscript has been modified according to the suggestion of the reviewer.

Round 2
Reviewer 2 Report
Thank you for your responses and clarifications to my comments. As already mentioned, the research investigates an important area of recycling of construction waste. However, the key deficiencies raised before about the research (mix) design, which are very critical and fundamental to the research process of this study, must be addressed before this work can be considered for publication. The detrimental effect of water absorption by aggregates on the water/cement ratio of concrete is well known in the concrete research industry, more especially in laboratory based experimental studies. Moisture content in aggregates (coarse and fine) must be considered and adjusted for when batching by weight in the laboratory study for accuracy of results.
Find below for your consideration my further comments on the issues raised:
Comment 1.
In response to the comments about how the high water absorption of RA was compensated for to avoid the mixing water being absorbed by RA, the authors provided an explanation in Lines 145 to 152 of the revised manuscript. The explanation provided is not satisfactory since adjustments were made in the mixes per m3 of concrete to cater for addition of different amounts of RCFRP fibers and RA. Furthermore, adjustment for water content in the mix to deal with high absorption of NA does not affect your selected idea of keeping the amount of RCFRP fibers and RA as the control variables and the amount of other ingredients staying constant. Both can be achieved in the same mix at the same time.
Also, the authors are trying to justify their lapse when coming up with the mix design by presenting that both RA and RCFRP fibers absorb water. This is misleading since RCFRP fibers have negligible water absorption compared to that of RA which was 4.18 times that of NA as presented in the manuscript. Furthermore, if for example you consider RA20RF5 mix in Table 3, it has 8.5 kg/m3 of RCFRP fibers compared to 216.32 kg/m3 of NA. Therefore, the magnitude of impact of NA in the mix in relation to water absorption is incomparable to that of RCFRP fibers.
As already mentioned before in previous comments, mixes should have had the same boundary conditions, and thus the W/C ratio should have been and can be maintained constant for all mixes in your study (even when you use your ‘idea’ of keeping the amount of RCFRP fibers and RA as the control variables and the amount of other ingredients staying constant). This is usually achieved in laboratory mixes by having the aggregates in saturated surface dry (SSD) moisture state to ensure that they will neither absorb not release water to the mix.
Comment 2.
In response to the comment on how 0% and 20% RA replacement rate selected to study the effect of RCFRP fibre contents, the authors explain in Lines 156 to 162 that 0% cannot facilitates the large-scale use of RA in construction and that too much addition of RA can affect the durability, mechanical properties and stability of RAC. They also provide research by Tam et al. [2] that recommend up to 20% of RA replacement rate for most applications including structural. This then raises the question why the authors selected RA volume replacement rates of 0%, 10%, 20%, 30% (in line 155) to be investigated in this study, if they knew beforehand that 10% and 30% RA replacement rate were not appropriate? Furthermore since RA volume replacement rates of 0%, 10%, 20%, 30% were already used in this study, it would be expected that the results obtained for the mechanical properties (compressive and flexural strengths) in this study would be used to select the RA replacement rate to study the effect of RCFRP fibre contents rather than basing on previous research. Past studies [2, 32] would be used to support their selection criteria.
Author Response
Thank you for your responses and clarifications to my comments. As already mentioned, the research investigates an important area of recycling of construction waste. However, the key deficiencies raised before about the research (mix) design, which are very critical and fundamental to the research process of this study, must be addressed before this work can be considered for publication. The detrimental effect of water absorption by aggregates on the water/cement ratio of concrete is well known in the concrete research industry, more especially in laboratory based experimental studies. Moisture content in aggregates (coarse and fine) must be considered and adjusted for when batching by weight in the laboratory study for accuracy of results.
Find below for your consideration my further comments on the issues raised:
Response: Thank you very much for the constructive and insightful comments. The authors are grateful for giving the opportunity to revise our manuscript. We have endeavored to improve the quality of the paper and clarify the comment on mix design.We agree with the reviewer that the mix design is critical for this study. The mix design of this study maintains the absolute water amount constant for all mixes. In this mix design, the actual water/cement ratio of concrete is different for mixes with different RA replacement rate. As discussed in the original manuscript, both the RA replacement rate and the change of water/cement ratio induced by RA addition can influence the mechanical properties of RAC. The authors agree that these two control variables make it difficult to isolate the influence of each control variable. Nevertheless, this study can still provide a useful insight into the synergetic effect of these two factors on the mechanical properties of RAC.Moreover, this study introduces an idea of using RCFRP fibers to improve the performance of RAC, therefore more attention is concentrated on the mechanical performance of mixes with/without RCFRP fibers. The results of these comparison are conducted based on the same RA replacement rate, which will not be influenced by the change in water/cement ratio. The authors appreciate the efforts of the reviewer in guiding us to improve the paper, the detailed responses of each comments are as follows:
Comment 1.In response to the comments about how the high water absorption of RA was compensated for to avoid the mixing water being absorbed by RA, the authors provided an explanation in Lines 145 to 152 of the revised manuscript. The explanation provided is not satisfactory since adjustments were made in the mixes per m3 of concrete to cater for addition of different amounts of RCFRP fibers and RA. Furthermore, adjustment for water content in the mix to deal with high absorption of NA does not affect your selected idea of keeping the amount of RCFRP fibers and RA as the control variables and the amount of other ingredients staying constant. Both can be achieved in the same mix at the same time.
Also, the authors are trying to justify their lapse when coming up with the mix design by presenting that both RA and RCFRP fibers absorb water. This is misleading since RCFRP fibers have negligible water absorption compared to that of RA which was 4.18 times that of NA as presented in the manuscript. Furthermore, if for example you consider RA20RF5 mix in Table 3, it has 8.5 kg/m3 of RCFRP fibers compared to 216.32 kg/m3 of NA. Therefore, the magnitude of impact of NA in the mix in relation to water absorption is incomparable to that of RCFRP fibers.
As already mentioned before in previous comments, mixes should have had the same boundary conditions, and thus the W/C ratio should have been and can be maintained constant for all mixes in your study (even when you use your ‘idea’ of keeping the amount of RCFRP fibers and RA as the control variables and the amount of other ingredients staying constant). This is usually achieved in laboratory mixes by having the aggregates in saturated surface dry (SSD) moisture state to ensure that they will neither absorb not release water to the mix.
Response 1: The authors appreciate the constructive comment of the reviewer. The authors agree that by adopting the aggregates in saturated surface dry (SSD) moisture state, the same water/cement ratio can be achieved. The mix design idea suggested by the reviewer is more conventional and can avoid the problem of high water absorption of RA. In this study, the absolute water amount of all mixes are kept constant, which cannot eliminate the influence of water/cement ratio change induced by RA addition. Nevertheless, the experiment of RA specimens with the same absolute water amount can still provide a useful insight into the synergetic effect of RA replacement rate and water absorption of RA on the mechanical properties of RAC. Moreover, this study introduces an idea of using RCFRP fibers to improve the performance of RAC, therefore more attention is concentrated on the mechanical performance of mixes with/without RCFRP fibers. These comparison are conducted based on the same RA replacement rate, which will not be influenced by the change in water/cement ratio. To avoid the misunderstanding from readers, the mix design in the revised manuscript is elaborated as follow:
This study maintains the absolute water amount constant for all mixes. Although the nominal water/cement ratios of different mixes are identical, the strong water absorption of RA can result in a lower actual water/cement ratio for mixes with high RA replacement rate. Even though the effect of water absorption of RA is not eliminated, the experiment of RA specimens with the same absolute water amount can still provide a useful insight into the synergetic effect of RA replacement rate and water absorption of RA on the mechanical properties of RAC. The RAC mix is treated as the reference mixes to investigate the mechanical performance of mixes with/without RCFRP fibers. The results of these comparison are conducted based on the same RA replacement rate, which will not be influenced by the water/cement ratio change induced by RA addition.
The above revision can be found in Lines 145 to 154.
Comment 2. In response to the comment on how 0% and 20% RA replacement rate selected to study the effect of RCFRP fiber contents, the authors explain in Lines 156 to 162 that 0% cannot facilitates the large-scale use of RA in construction and that too much addition of RA can affect the durability, mechanical properties and stability of RAC. They also provide research by Tam et al. [2] that recommend up to 20% of RA replacement rate for most applications including structural. This then raises the question why the authors selected RA volume replacement rates of 0%, 10%, 20%, 30% (in line 155) to be investigated in this study, if they knew beforehand that 10% and 30% RA replacement rate were not appropriate? Furthermore since RA volume replacement rates of 0%, 10%, 20%, 30% were already used in this study, it would be expected that the results obtained for the mechanical properties (compressive and flexural strengths) in this study would be used to select the RA replacement rate to study the effect of RCFRP fibre contents rather than basing on previous research. Past studies [2, 32] would be used to support their selection criteria.
Response 2: The authors appreciate the constructive comment of the reviewer. In fact, both previous studies and the experiment results of this study are adopted to guide the selection of representative RA replacement rate. As shown in Figures 7 and 11, the RAC mix with 30% RA replacement exhibits relatively large strength reduction compared with the mixes with 10% and 20% RA replacement rate. According to the suggestion of the reviewer, the reason on the selection of RA replacement rate is revised as follow:
In this study, RA volume replacement rates of 0%, 10%, 20%, 30% are investigated. Considering that, 10% of RA replacement rate is small, which cannot facilitates the large-scale use RA in construction. On the other hand, RA has the characteristics of low density, low wear resistance and compressive strength, high water absorption and porosity [32], too much addition of RA may affect the durability, mechanical properties and stability of RAC. According to the pre-experiment of RAC with 10–30% replacement rate in this study, the RAC mix with 30% RA replacement exhibits relatively large strength reduction compared with the mixes with 10% and 20% RA replacement rates, which is similar to the founding of Tam et al. [2]. Therefore, the mix with RA replacement rate of 20% is selected to further investigate the synergetic effect with RCFRP fibers.
The above revision can be found in Lines 157 to 166.

Round 3
Reviewer 2 Report
Thank you for your responses and clarifications to my comments. It is true that this study may still provide a useful insight into the synergetic effect of RA replacement rate and water absorption of RA on the mechanical properties of RAC. However, the influence of water/cement ratio change induced by RA addition should be highlighted and discussed in the manuscript. Below are the minor comments to guide you in addressing this aspect.
- For the revision provided in Lines 145 to 154, since the absolute water amount of all mixes being kept constant cannot eliminate the influence of water/cement ratio change induced by RA addition, provide a brief discussion of how the change of water/cement ratio induced by RA addition can influence the mechanical properties of RAC.
- For the revision provided in Lines 157 to 166, clearly provide in the manuscript the selection criteria (as done in your response to my comment) by mentioning the observations in Figures 7 (a), 11(a) and even 13 (a) where the RAC mix with 30% RA replacement exhibits relatively large strength and stiffness reduction compared with the mixes with 10% and 20% RA replacement rate. Also revise and improve your grammar in the revised section. For example, the sentence in Lines 158 to 159 (“Considering that, 10% of RA replacement rate is small, which cannot facilitates the large-scale use RA in construction”) is incomplete and lacks clarity. Also the sentence in Line 162 “According to the pre-experiment of RAC with 10–30% replacement rate in this study” can be improved to “According to the initial study conducted on RAC when the replacement rate was varied between 10–30% as shown in Figures 7 (a), 11(a) and 13 (a), the RAC mix with 30% RA replacement exhibited.............”
Author Response
Thank you for your responses and clarifications to my comments. It is true that this study may still provide a useful insight into the synergetic effect of RA replacement rate and water absorption of RA on the mechanical properties of RAC. However, the influence of water/cement ratio change induced by RA addition should be highlighted and discussed in the manuscript. Below are the minor comments to guide you in addressing this aspect.
Response: Thank you very much for the constructive and valuable comments. The authors are grateful for giving the opportunity to revise our manuscript. The paper has been revised according to the comments of the reviewer and detailed responses are as follows.
Comment 1. For the revision provided in Lines 145 to 154, since the absolute water amount of all mixes being kept constant cannot eliminate the influence of water/cement ratio change induced by RA addition, provide a brief discussion of how the change of water/cement ratio induced by RA addition can influence the mechanical properties of RAC.
Response 1: The authors appreciate the constructive comment of the reviewer. According to the work of Poon et al. [34], if the recycled aggregate is used in the oven-dried state, water may move from the bulk cement matrix toward the recycled aggregate, which can result in a slightly higher compressive strength compared with that of the case with the recycled aggregate in saturated surface-dried state.
The above revision can be found in Lines 151 to 155.
[34] Poon, C.S.; Shui, Z.H.; Lam, L.; Fok, H.; Kou, S.C. Influence of moisture states of natural and recycled aggregates on the slump and compressive strength of concrete. Cem. Concr. Res. 2004, 34, 31–36. doi:10.1016/S0008-8846(03)00186-8.
Comment 2. For the revision provided in Lines 157 to 166, clearly provide in the manuscript the selection criteria (as done in your response to my comment) by mentioning the observations in Figures 7 (a), 11(a) and even 13 (a) where the RAC mix with 30% RA replacement exhibits relatively large strength and stiffness reduction compared with the mixes with 10% and 20% RA replacement rate. Also revise and improve your grammar in the revised section. For example, the sentence in Lines 158 to 159 (“Considering that, 10% of RA replacement rate is small, which cannot facilitates the large-scale use RA in construction”) is incomplete and lacks clarity. Also the sentence in Line 162 “According to the pre-experiment of RAC with 10–30% replacement rate in this study” can be improved to “According to the initial study conducted on RAC when the replacement rate was varied between 10–30% as shown in Figures 7 (a), 11(a) and 13 (a), the RAC mix with 30% RA replacement exhibited.............”
Response 2: The authors appreciate the constructive comment of the reviewer. The manuscript has been revised as follow:
In this study, RAC mixes with the aggregate replacement rates of 0%, 10%, 20%, 30% are investigated. To further study the performance of RAC with RCFRP fibers, the representative RA replacement rate of 20% is selected. This is because 10% RA replacement rate is relatively small, the effect of saving natural aggregate is not obvious in this replacement rate. Moreover, RA has the characteristics of low density, low wear resistance and compressive strength, high water absorption and porosity [35], too much addition of RA may affect the durability, mechanical properties and stability of RAC. According to the initial study conducted on RAC when the replacement rate was varied between 10–30% as shown in Figures 7a, 11a and 13a, the RAC mix with 30% RA replacement exhibits relatively large strength reduction compared with the mixes with 10% and 20% RA replacement rates. Therefore, the mix with RA replacement rate of 20% is selected to further investigate the synergetic effect of RAC with RCFRP fibers.
The above revision can be found in Lines 161 to 172.
Once again, the authors want to thank the reviewer for the valuable and constructive comments and suggestions which helped to improve the manuscript.
